# The contribution of birth plans to shared decision-making from the perspectives of women, their partners and their healthcare providers

Naaz Shareef[1]*, Poshya Said[1]☯, Silke Lamers[1]☯, Marianne Nieuwenhuijze[2,3], Marieke de Vries[4], Jeroen van Dillen[5]

1 Faculty of Medical Sciences, Radboud University Medical Center, Nijmegen, The Netherlands, 2 Research Centre for Midwifery Science, Zuyd University, Universiteitssingel 60, Maastricht, The Netherlands, 3 CAPHRI, Maastricht University, Universiteitssingel 60, Maastricht, The Netherlands, 4 iCIS, Radboud University, Nijmegen, The Netherlands, 5 Department of Obstetrics, Radboud University Medical Center, Nijmegen, Netherlands

☯ These authors contributed equally to this work.
* naaz.shareef@radboudumc.nl

**Data Availability Statement:** Data cannot be shared publicly because of the sensitive and confidential nature of the research material. The research is based on interviews that involve

## Abstract

### Background

The birth plan is a document expressing a pregnant woman's childbirth preferences, enabling communication of expectations and facilitating discussions among women, their partners, and healthcare providers for key birthing decisions. There has been limited research on the role of birth plans in shared decision-making (SDM). Our study aims to explore how the use of birth plans can contribute to SDM from women's, partners, and healthcare providers' perspectives.

### Methods

We conducted in-depth interviews with women, their partners, and their healthcare providers. We used a thematic analysis to identify themes and subthemes. Furthermore, we created a grounded theory about the role of birth plans as a tool in SDM.

### Results

Three main themes were created: *"Creating a birth plan"*, *"Getting all on board"* and *"Birth plans in the daily practice of decision-making"*. Most women, partners, and healthcare providers agreed that birth plans can facilitate communication and SDM. Women and their partners viewed the birth plan as a tool to prepare for birth. Most healthcare providers mentioned the birth plan as a tool to get to know the women, their partners, and their preferences. Barriers are the attitude of healthcare providers toward birth plans, such as their evident resistance to the birth plan itself or to certain preferences. Another barrier is the assumption women and their partners may have that these plans can accurately predict the

women, partners, and healthcare providers, all containing sensitive personal information, highlighting the need for privacy measures. Participants only agreed to these interviews with the condition of anonymity, and thus, the authors are forbidden by them from making the interview content fully public. Selected parts of these transcripts may be provided to researchers who meet specific criteria, after a request to the medical ethics committee of Radboud University at commissiemensgebondenonderzoek@radboudumc.nl. This process ensures the authors abide by the agreements with the participants and the Medical Ethics Review Committee (METC), and honor their data management and ethical standards, particularly regarding the challenge of maintaining anonymity in such studies.

**Funding:** The author(s) received no specific funding for this work.

**Competing interests:** The authors have declared that no competing interests exist.

childbirth experience, enhancing the chance of a disappointing, negative experience. Some healthcare providers view birth plans as barriers to SDM.

## Conclusion

The use of a birth plan seems to promote women's, partners', and healthcare providers' involvement in the birth process, and seems suitable to facilitate SDM. Further research is required to explore strategies for overcoming barriers, including healthcare providers' attitudes toward birth plans and the expectations of women and their partners regarding their role.

## Introduction

Shared Decision-Making (SDM) in healthcare is a collaborative process in which patients and their healthcare providers actively participate in making decisions regarding care [1]. SDM typically involves several stages [2]. The first stage is "choice talk," during which a choice and the need for decision-making are introduced. Once the patient is aware of the need to make a choice, the second step, "option talk," is introduced. This stage involves deliberation, where the healthcare provider and the patient discuss the available options, considering the patient's preferences, values, and goals. The following stage is "decision talk," during which the patient reflects on their values and priorities to make an informed decision, ultimately leading to a choice. Throughout these stages, effective communication, active engagement, and mutual respect are vital for achieving SDM [1,3].

There is an increasing desire among pregnant women and their partners to actively participate in decision-making throughout pregnancy, childbirth, and postpartum [4]. Decision-making enhances personal control and self-efficacy and contributes to a more positive childbirth experience [5,6]. In the Netherlands, the maternity care system is known for its physiological approach to pregnancy and childbirth, offering primary care to low-risk women and secondary care and tertiary care for higher-risk pregnancies. This system provides options for home or outpatient hospital births for low-risk women and hospital births for those in secondary care. It is all underpinned by the 2016 Integrated Maternity Care Standard, which recommends individual birth care plans for every pregnant woman [7]. Within maternity care, birth plans can be used to stimulate SDM [8]. Typically, a birth plan is a written document by a pregnant woman and her partner during pregnancy, outlining their preferences for childbirth [8]. It serves as a means for pregnant women to express their expectations, preferences, and values regarding childbirth and can facilitate meaningful conversations among women, partners, and healthcare providers regarding decision-making [8,9]. Creating a birth plan offers the opportunity to provide women and their partners with more awareness about the process of childbirth by engaging in discussions with healthcare providers regarding their knowledge and various possible scenarios [10].

A birth plan has several aims. The first aim is to enhance education before childbirth [8]. Throughout pregnancy, pregnant women and their partners gain a deeper understanding of their options, making them feel more confident and prepared for unexpected decisions during childbirth when time is limited [8,11,12]. The second aim of a birth plan is to assist women and their partners in developing informed preferences based on the educational information they have obtained regarding medical interventions like labor induction or epidurals [8,13,14]. These preferences are subsequently documented in the birth plan. The third objective is to

foster effective communication [8,13,15,16]. The birth plan serves as a tool for women and healthcare providers to understand each other's perspectives better. It is created and discussed with healthcare providers during pregnancy, ensuring that the providers are aware of the woman's values and can anticipate any unrealistic expectations [8,13,17]. The fourth aim of utilizing a birth plan is to empower women to exert greater autonomy during childbirth if desired [8,12]. The process of writing and discussing a birth plan allows women to feel in control of aligning their birthing process with their preferences, thereby promoting a sense of empowerment [11,15,18].

There has been limited research conducted on the role of the birth plan in SDM [8]. Considering the significance of SDM and the potential influence of birth plans, further research is essential to better understand their effectiveness and implementation.Our study adds value by investigating the perspectives of women, partners, and healthcare providers on the role of birth plans for SDM in maternity care.

## Methods

This study is an explorative, qualitative interview study. Given the absence of a theory explaining the impact of birth plans on SDM, we utilized the grounded theory approach to guide our data collection and analysis. In our research, we combined data from interviews concerning birth plans in relation to SDM with three distinct groups–women, partners, and healthcare providers–, all conducted within the framework of one study. These interviews were performed at the Radboud University Medical Center (Radboudumc) in Nijmegen, the Netherlands. This is an academic teaching hospital providing secondary care for the local community and tertiary care for higher-risk pregnancies for the larger region. This hospital is committed to person-oriented care and the first in the Netherlands to endorse the Salzburg Declaration for shared decision-making [19].

### Participants

The population of interest were women who used a birth plan, partners of women who used a birth plan and different healthcare providers in maternity care.

**Women.**   For interviews with women, 178 women who recently gave birth in Radboudumc were approached for the interviews (Nsh and Nsc) from March to April 2021 during the standard telephone follow-up three weeks postpartum (Fig 1). Women with a birth plan were asked to participate in an interview study. All women who were invited received an information and informed consent letter via e-mail and after agreement and signing, they were included in the study. Inclusion criteria were women who had a birth plan and women who recently gave birth in Radboudumc (1-3months postpartum). Exclusion criteria for the women's interviews were maternal and perinatal morbidity or mortality and language barrier. We aimed to ensure the group was as diverse as possible, so we included multiple categories such as different ages, gestational age at labor, gravida/para status, mode of birth, and medical indications. Additionally, these women have no relationship with the men we interviewed or the healthcare providers.

**Partners.**   For interviews with partners (Nsh and SL), 217 women who gave birth in the Radboudumc were approached from December 2021 till February 2022 (Fig 2). The men we interviewed were recruited during follow-up telephone conversations 1–3 months postpartum with other women who had given birth. These women gave permission for us to approach their partners for further research. All partners who were invited for an interview, received an information and informed consent letter via e-mail. Only after agreeing with the consent form were they included in the study. Inclusion criteria for partners were: that their women had a

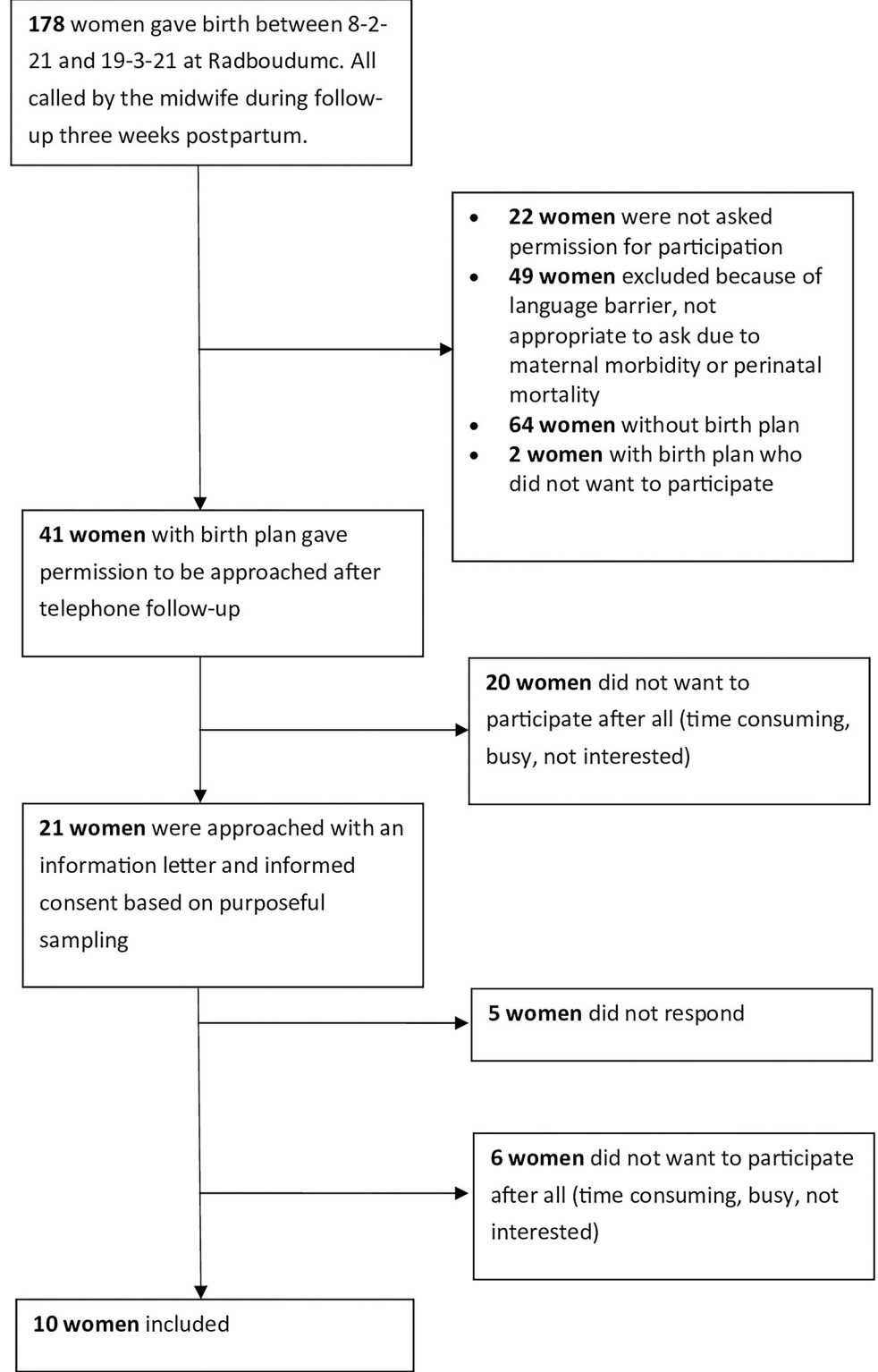

**Fig 1. Selection process of the women interviewed.**

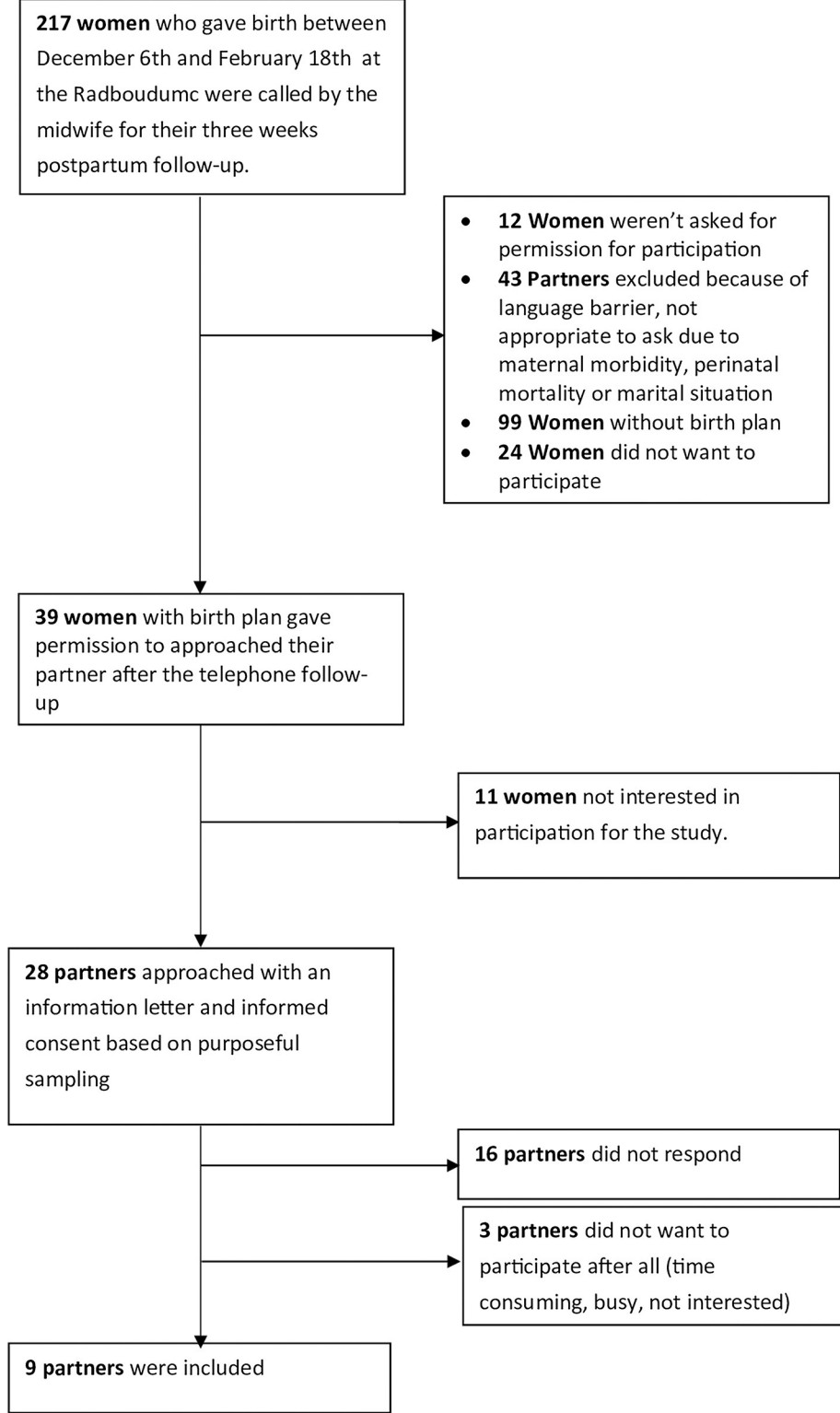

**Fig 2. Selection process of the partners interviewed.**

birth plan and that they were partners who recently gave birth in Radboudumc (1–3 months postpartum). Exclusion criteria for the partner's interviews were maternal and perinatal morbidity or mortality and language barrier. These partners had no relationship with the women we interviewed or with the healthcare providers. We examined characteristics of both the men, such as age, and their women, involving individuals with diverse characteristics to ensure a broad representation.

**Healthcare providers.** Healthcare providers were contacted via email by JvD, who requested them to respond if they were willing to participate, and subsequently received an invitation for an interview from Nsh and PS, from February to May 2022. These providers, located in the Nijmegen region, were approached via email with a request for participation. To ensure a broad representation, we deliberately formed a diverse group by selecting healthcare providers based on their function (e.g., gynecologist, midwife), gender, and their workplace setting (primary or secondary care). The healthcare providers involved in these interviews had no prior relationship with the women or partners participating in the study. We have not provided extensive details about the healthcare providers to maintain anonymity and protect privacy, as many providers in this region are well-acquainted with each other.

## Data collection

Due to COVID-19, all interviews were held via the digital platform Microsoft Teams.

In our interviews, we used open-ended questions that explored the experience and impact of birth plans on SDM, such as the impact on communication, experience, and involvement. A topic list (see S1–S3 Tables) was used for the interviews and was adjusted, if necessary, after each interview.

The duration of the interviews was between 30 minutes and 90 minutes. A member check was used after conducting the in-depth interviews to ensure the information in the interviews was correct and to check if changes were needed. No changes were made after the member check.

## Data analysis

An alternation of data collection and data analysis was performed to improve the quality of the interviews. NSc and NSh coded the transcripts for the interviews with women, SL and NSh for the interviews with partners and PS and NSh for the interviews with healthcare providers. Six transcripts (60%), the first and last three transcripts, were double-coded. This collaboration in coding allowed interviews to be viewed from different angles, providing fewer assumptions and ensuring more reliability. We tried to collect data until the properties of our theoretical categories were saturated, meaning fresh data no longer sparked new theoretical insights. Data saturation was achieved after eight (of ten) interviews with women and ten (of 13) with healthcare providers, which was confirmed after analyzing the two remaining interviews per category. After nine interviews with partners, data saturation seemed to occur. However, due to a lack of time, no more interviews were conducted for confirmation of saturation. The software program ATLAS.ti (version 8.4.20) was used to assist with coding.

The analytical process began with close reading of the transcripts. Open, axial and selective coding was performed to fragment the data into codes, categories, themes and eventually determine an overarching theme. Data was reorganized and coded at multiple levels, using coding and constant comparison, within and between interviews. The code tree, drawn up by NSh, SL and PS can be found in Fig 3. Quotes illustrate the findings. These quotes were translated forwards and backwards from Dutch to English and vice versa.

### Societal and ethical justification

Permission for this study was requested from the regional medical ethical committee of the Radboud University Nijmegen (women file number 2021–7334, partners file no. 2022–1353, care providers file no. 2022–13534). The committee considered this study as not requiring ethical permission for its implementation.

All data were anonymized and safely stored in a secured place in the network at the Radboudumc, only accessible for the research team. Before inclusion, participants signed for informed consent and gave permission for their quotes to be used.

Consolidated criteria for reporting qualitative studies (COREQ): 32-item checklist has been used to report the methods [20].

### The research team and reflexivity

Three Radboud University medical students, NSc, SL, and PS, with initial supervision from JvD, conducted interviews with women, their partners, and healthcare providers, respectively. JvD, NSh, MN, and MdV contributed to article revisions, while JvD, MH, and MN, experts in midwifery/obstetrics and SDM, guided the research.

## Results

Ten interviews were conducted with women, nine with partners, and thirteen with healthcare providers. All the women who were included have given birth at Radboudumc, varying

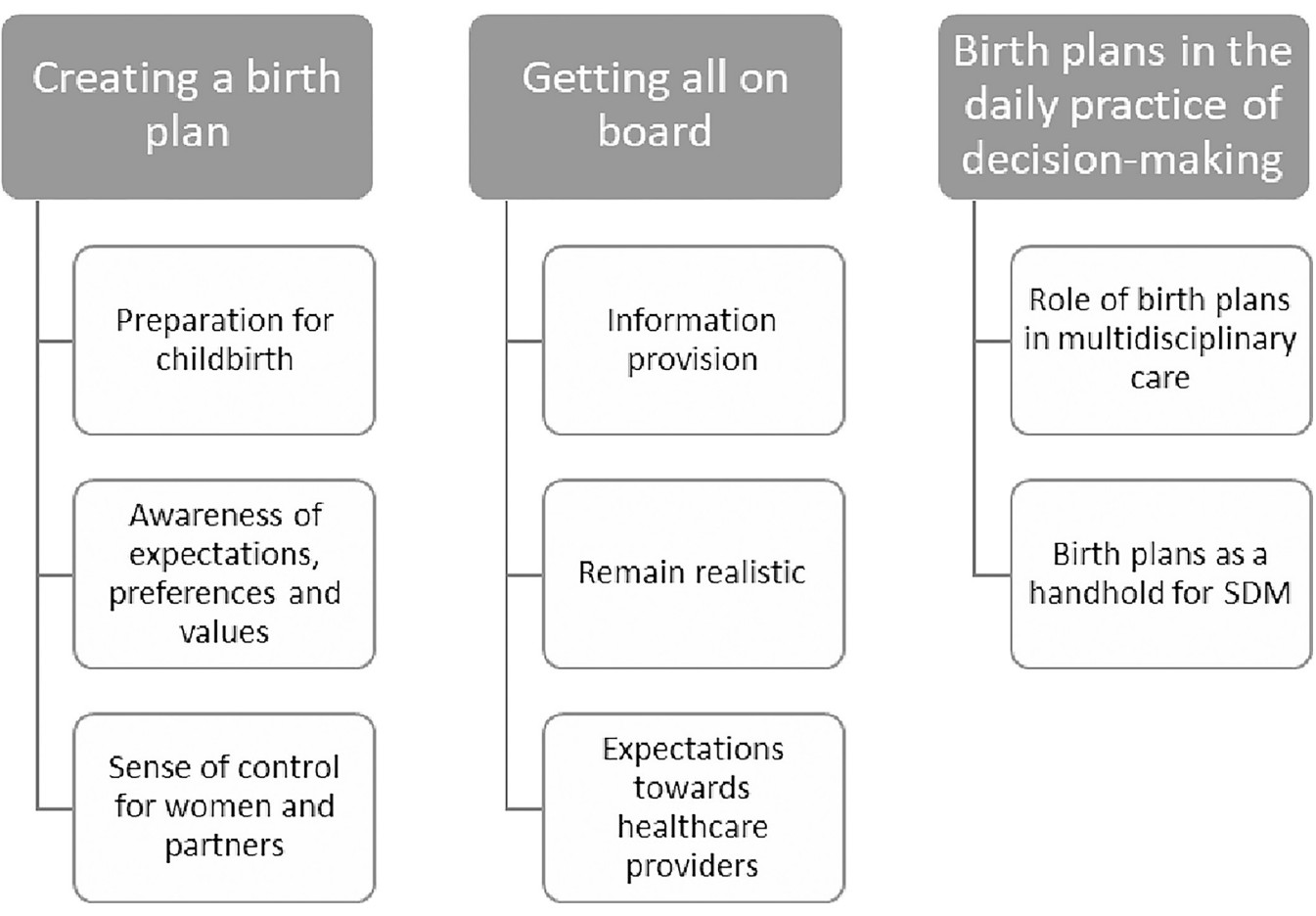

**Fig 3. Themes and subthemes.**

between primiparous and multiparous. The interviewed partners consisted exclusively of males whose spouses recently gave birth at the Radboudumc. The healthcare providers were predominantly female, with the exception of one, alternating between midwives, gynecologists, or gynecologists in training. These providers operated in both primary and secondary care settings. Further characteristics of the interviewed participants are shown in S4–S6 Tables.

After thematic analysis of the in-depth interviews, we found that three main themes were important: *"Creating a birth plan"*, *"Getting all on board"* and *"Birth plans in the daily practice of decision-making"*, as illustrated in Fig 3. Based on the collected data we generated a theory, which we present under the heading 'The Grounded Theory: birth plans as a tool for SDM.' Our study shows that several elements are important when using a birth plan to stimulate SDM.

## Theme 1: Creating a birth plan

The first theme focused on the conditions needed to create a birth plan and promote SDM. This theme identified three sub-themes: *preparation for childbirth*, *awareness of expectations, preferences and values*, and *sense of control for women and partners*. Preparation and awareness seemed to be meaningful for all stakeholders. The *sense of control for women and partners* is a theme that was primarily discussed by women and their partners.

**1.1 Preparation for childbirth.** Preparation for childbirth was frequently discussed during the interviews with both women and partners. Women appreciated their partners' involvement in the process both before and during childbirth. According to the women and partners, the use of a birth plan facilitates this by enabling joint decision-making based on the shared knowledge they have acquired.

The approach to childbirth preparation among the participating women and partners appeared to vary. Women mentioned that a birth plan prompted them to contemplate their expectations, preferences, and values regarding childbirth, aiding in decision-making. Since partners collaborated with the women to construct the birth plans, they too were motivated to ponder their expectations, possibilities, and strategies for assisting women during labor.

According to partners in this study, engaging in preparation through the birth plan ensured heightened partner participation throughout childbirth. Moreover, due to their increased level of information and readiness, partners perceived to be more actively engaged in the decision-making process.

However, a recurring aspect that also emerged from the interviews with partners was that while creating the birth plan, partners often deferred decisions primarily to the women, considering birth as a process inherent to the female body.

> *"Yes, my girlfriend wanted that and we talked about it. But it's her body, so it's her choice. This was the same mentality I had while drawing up the birth plan, if my girlfriend wants something I will always go along with it. I think that it is very crazy and even a little selfish if I had an opinion about the way of delivery. Because it's just not my body and the baby isn't coming out of me. I also said to my girlfriend that if she initiates the decisions, there is a 99% chance that I will approve."* (Participant 6-Partner)

Another crucial topic based on the experiences of healthcare providers with preparation for childbirth, was maintaining realistic expectations, which will be further elaborated in the sub-theme *remain realistic*.

**1.2 Awareness of expectations, preferences and values.** The following subtheme revolves around the awareness of expectations, preferences, and values. Creating a birth plan has the

potential to enhance a woman's and her partner's awareness regarding their preferences and expectations related to childbirth. Furthermore, collaborative birth plan creation can prompt couples to engage in discussions about their preferences. This, in turn, gives the opportunity to facilitate an understanding of each other's desires and requirements. Moreover, women highlighted that they had gained insight into their partner's expectations, fostering greater partner engagement in the process due to their preparedness for requisite decisions.

*"For us it was also a kind of discussion paper, it helped us to talk about what was going on: what are the pros, what are the cons, what do we actually want and what do we not want. This way we were also better aware of the other's wishes." (Participant 8-Partner)*

The birth plan can also be a way for the healthcare provider to start a conversation and discover the woman's values. Several healthcare providers implied that this helped them to implement patient-centered care and support the woman to make her own decisions. If the vision of the mother and partner presented medical risks, they felt like it was their duty to discuss these risks. This allowed the healthcare provider to ensure safety for both the mother and baby.

*"So if there are very unrealistic things or certain expectations that are very high on their birth plans, which may not be achievable or present medical risks, I find it good to tackle this beforehand. That you understand them and their vision, but that childbirth is not plannable and that people are aware of this." (Participant 6- Healthcare provider)*

**1.3 Sense of control for women and partners.** Many participants expressed their intent to outline their expectations, preferences, and values within a birth plan to safeguard their autonomy and foster a feeling of control. This is particularly crucial given that expectant women are often uncertain about the circumstances they will face during childbirth. During the interviews, the significance of autonomy emerged prominently. Multiple stakeholders, both women and partners, emphasized that having a sense of autonomy allowed them to be active participants in the process and facilitated their involvement in decision-making.

*"I wanted a birth plan because during my first birth, I didn't have control over the birth of my child. Now I wanted to be prepared for myself so I could be more relaxed during my pregnancy and birth. The healthcare provider asked if I had a birth plan and she assured me that they will try their best. This gave me more confidence and made me feel relaxed and I had a feeling that I was more in control now over my own birth. "(Participant 6- Woman)*

Furthermore, an important aspect of sense of control is that women stated that the birth plan can empower the partner to function as an advocate for the woman's autonomy. According to numerous women and their partners, assuming this role of advocate holds significant importance during childbirth. The partner is entrusted with safeguarding the mother's values and advocating on her behalf if she cannot do so during the birthing process.

*"You are aware of what's written down in the birth plan and we know each other's wishes. For this reason, I was able to talk about X's (name of partner) wishes with the doctors when she was having contractions. Yes, in those moments you really have to be the one in charge, because she has to be concentrated on giving birth. In addition, I'm there to indicate things now and then and to guide X (name of partner) every now and then in making decisions."(Participant 6- Partner)*

This sub-theme was mentioned in particular by women and partners. Still, the healthcare providers also indicated that the woman's autonomy is considered very important and that they try to stimulate the woman in this regard.

## Theme 2: Getting all on board

The subsequent theme centers on the conditions needed to use birth plans for SDM. The creation of a birth plan necessitates the *provision of information* to expectant mothers and their partners. In this process, healthcare providers play a major role as they are responsible for delivering accurate information and ensuring that women and their partners can make informed decisions while *remaining realistic*.

Lastly, the mutual *expectations* associated with both each other and the birth plan will be examined. These subthemes, primarily brought up by women and healthcare providers, underscore the significance of these factors for the effective functionality of a birth plan.

**2.1 Information provision.** A frequently mentioned aspect regarding the theme *"getting all on board"* is providing information to women and their partners around childbirth. According to healthcare providers, birth plans provide healthcare providers with essential information about the woman, enabling them to understand her desires and expectations. This understanding guides them in making shared decisions and ensures that the woman's preferences are respected. Most acknowledged that ultimately women have the final say in the decision-making process because it is their unique journey of childbirth.

> *"Look, you don't know the frame of reference and preferences of the patient if you do not have explicit conversations about it. A birth plan is a beautiful tool for such a dialogue. Without it, it is also possible, but this does certainly help with making decisions." (Participant 2- Healthcare provider)*

The pregnant women and their partners mentioned that a birth plan was seen as a tool for dialogue in which the healthcare provider could offer information and advice and exchange ideas with the women and her partner, if applicable. The conversation can clarify the woman's preferences and fears, which enables the healthcare provider to understand what matters most from the woman's perspective and tailor care to that.

Besides the positive aspects of a birth plan as a tool to facilitate SDM and provide guidance, it can also be seen as a barrier, resulting in challenging communication. From the healthcare providers' perspective, this happens with complex and/or comprehensive birth plans when SDM is difficult to facilitate, because there is no space for discussing other scenarios. As stated above, according to the healthcare providers it is important to take into account that a birth plan is a guideline, not a contract. These types of birth plans can lead to challenges in communication and therefore are seen as barriers by some healthcare providers due to extra pressure that comes with the complexity of the plans and results in less room for conversation and/or discussion, which is fundamental in facilitating SDM.

> *"But if someone creates a plan, puts it in front of you and says: this is how I want it, and not different. That is a lot different from when there is room for conversation and consultation. If this is the case, there is no shared decision-making to facilitate. It is a one-way street."*
>
> *(Participant 10- Healthcare provider)*

According to some partners, a barrier is that care providers occasionally provide too much information. Especially during childbirth, this can be experienced as aggravating. A woman

and her partner need to be able to focus on the delivery, therefore absorbing all information can be seen as a struggle and making decisions could be difficult. According to several partners, the healthcare provider must ensure that she/he provides sufficient information to promote shared decision-making, but that partners are not overloaded.

> "*I did notice that X (name of partner), when she was in a lot of pain and very tired, she wanted limited information. This while the midwife and nurses want to inform you as well as possible about every decision. For example, the moment you want to take an epidural you get a whole list with comments and possible consequences, while at that moment you have so much pain which leads to the fact that all the information goes in one ear and out the other. First, you need to adopt all the information from the obstetrician, then again from the nurses and then from the anesthetist. I understand that it's protocol, but all that information is gone before you know it.*" (Participant 8- Partner)

Nonetheless, a disparity in the requirement for guidance from healthcare providers emerged between primiparous and multiparous women. Several primiparous women noted the challenges in formulating a birth plan due to their limited familiarity with potential options and childbirth scenarios. This group favored acquiring extensive knowledge owing to their absence of prior experiences, which in turn empowered them to make well-informed decisions. Conversely, multiparous women possessed a clearer understanding of their preferences during labor, drawn from their prior childbirth experiences, enabling them to more confidently discern their preferences.

> "*Of course, with a first birth, you don't really know how you are doing, whether you are there, or whether you are under anesthesia. So, for me it was a feeling of security creating a birth plan, like, suppose I'm not doing well, or my partner can't handle it all, then at least it's fixed in the birth plan*" (Participant 1- Woman).

**2.2 Remain realistic.**   Multiple healthcare providers believe it is important to discuss the pros and cons with women about their expectations, preferences and values and to communicate clearly about different options and scenarios. Expectation management is frequently named as an important factor in SDM and realizing a birth plan. The interviews revealed a consensus among stakeholders about the importance of discussing birth preferences. Simultaneously, they emphasized the need for realism in decision-making, acknowledging that not all options may be viable or advisable due to potential risks for the woman or her baby.

Pregnant women can see the birth plan as a preparation for birth, which makes it important to stay realistic in their expectations. Just as pregnant women find it important to discuss the possibilities, their partners also mentioned realism as an important aspect to the birth plan because childbirth is largely unplannable and is an important point of discussion to prevent disappointments.

The healthcare providers suggested that expectation management prevents false expectations, trauma, and eventually negative birth experiences. They implied that with the use of birth plans, these expectation issues can be tackled early on to prevent discussions during or right before childbirth. In this way, women and their partners know what to expect during childbirth and prepare mentally for the childbirth, to prevent trauma.

> "*We discuss these kinds of things together to see if we are on the same page and what your wishes are and if that is all possible here. And to figure out what we can do for you, or what*

*we simply cannot do because we do not have the right resources. We will go over it and calmly discuss it the next time. I always tell them to try to think clearly and to consider that a birth plan does not always go the way you want it to go. You can write down many wishes, but this can also be very disappointing eventually, which you have to realize with everything you may wish. That some things do not go the way you want. That is expectation management."* (Participant 5- Healthcare provider)

Therefore, according to women, it is advisable for partners and healthcare providers to engage in discussions concerning potential childbirth scenarios and the decisions that women and their partners might encounter during labor. It's worth acknowledging that the actual birthing process might deviate from expectations, underscoring the importance of understanding that a birth plan may not perfectly align with reality. Pregnant women suggested that this aspect could be highlighted more explicitly at times, as it could potentially reduce instances of disappointment and foster smoother communication.

**2.3 Expectations towards healthcare providers.**   Women highlighted their expectation of having choices and underscored the importance of healthcare providers adopting an approach that is open, impartial, and practical. They suggested that providers should present the advantages and disadvantages of various options while also offering medical insights to guide women and partners towards informed and secure decisions.

Partners demonstrated substantial confidence in healthcare providers' decision-making capabilities, driven by the belief that providers would prioritize their best interests, especially in unforeseen circumstances.

"*We mainly left it to the experts, when things didn't turn out the way we expected. And we actually trusted them a lot, by thinking if something happened they will do what is best for us.*" (Participant 1- Partner)

**2.4 Expectations towards the function of birth plans.**   When faced with complex and lengthy birth plans, healthcare providers often mentioned a sense of falling short. Healthcare providers frequently reported feeling inadequate when dealing with complex and extensive birth plans, as the difficulty in meeting all the specified preferences, some potentially unachievable, added to the pressure of their responsibilities. However, they still play a crucial part in collaborating with the woman and her partner to make shared decisions regarding a treatment plan that aims for a positive birth experience. Healthcare providers mentioned that it is important to recognize that, ultimately, the woman has the final decision-making authority, even though healthcare providers may feel like they have failed her.

Most interviews with healthcare providers revealed that they have encountered opposition concerning the term 'birth plan' because it can foster unrealistic anticipations among pregnant women. Additionally, women highlighted that childbirth itself might not be entirely predictable. Women mention that the idea that childbirth could be 'plannable' because of the name 'birth plan' could potentially complicate the communication between healthcare providers and women, posing challenges to facilitating SDM.

"*Healthcare providers should mention that a birth plan is more an indication of how you want your birth to go, but they should emphasize that birth is unpredictable and can go differently than you want. If healthcare providers only mention that sentence, that would be enough for me to realize that birth can go differently than planned and I wouldn't mind if things go differently than planned, because I know this beforehand*"(Participant 2- Woman)

### Theme 3: Birth plans in the daily practice of decision-making

Once the birth plan has been formulated and deliberated upon, its execution within the daily clinical practice becomes necessary. The subsequent theme is focused on what *the role of birth plans in multidisciplinary care* is and how it can be used as *a handhold for SDM.*

**3.1 Role of birth plans in multidisciplinary care.**   The woman's expectations, preferences, and values hold significant importance for all healthcare providers engaged in the childbirth process. For instance, one participant (a woman) incorporated a consultation with an anesthetist regarding pain relief options in her birth plan, aiming to experience labor with tranquility. Numerous women and partners noted that the integration of birth plans within multidisciplinary care enhances SDM by encompassing various specialties and promoting patient-centered care.

By involving consultations with diverse healthcare professionals such as anesthetists for pain relief choices, birth plans potentially empower women to make well-informed decisions, fostering a composed and confident approach to labor. This collaborative methodology nurtures a comprehensive and holistic approach to decision-making, ensuring women's preferences and requirements are incorporated across a spectrum of care aspects.

> "*I think a birth plan is a very handy tool for shared decision making because in this way, it is clear for everyone involved in labor what you discussed with the gynecologist and what your wishes are [. . .] I also had two consultations with the* anaesthetist, *to discuss the possibilities, but also about the scenarios of a cesarean section. About epidural, spinal and also anesthetic. I knew more, so I could also make more choices. I could also clearly indicate what I did and did not want in my next labor.*" *(Participant 10- Woman)*

In the interviews, the healthcare providers also mentioned that the use of birth plans can also lead to improvements in multidisciplinary care. As a result, values written down in the birth plan can be communicated more easily with other healthcare providers. Hence, the transfer of patients from primary care to secondary care is often easier. However, a difference in attitude was seen in primary and secondary care healthcare providers. The healthcare providers mentioned that the use of birth plans in primary care is standard and more automatized and implemented than in secondary care.

> "*Yes, what we were just talking about if it was not fully discussed in primary care and there are things in it, or there are wishes that are not. . . If indeed the woman's wish is to stay as long as possible waiting while we think yes.. You have actually been doing that for a long time at home. And erm.. It is now time to do something, then it can indeed be an obstructing factor. Anyway.. That's what we just talked about a little bit. In principle, I think it's very nice that if you don't know people you can make a bit of an assessment of their position. And uh. . . So then. . . Then it is precisely in those situations. . . It is very helpful.*" *(Participant 1- Healthcare provider)*

**3.2 Birth plans as a handhold for SDM.**   In general, women and their partners expressed positivity towards the birth plan as it offered them a handhold to prepare for childbirth, decision making and it served as a point of reference whenever needed.

> "*Using a birth plan helped with shared decision-making, because my wishes in the birth plans were discussed several times. My opinions about several topics about birth were already known and my own midwife was also present to vouch for my birth plan wishes. She was the*

*one discussing my birth plan during birth and she knew exactly what I would want to choose in that moment, so using the birth plan really worked with making decisions." (Participant 3-Woman)*

Healthcare providers were generally positive about the birth plan as well. According to them, the birth plan serves as a valuable instrument for fostering shared decision-making by facilitating dialogues between healthcare providers and women (as well as their partners), thereby enhancing comprehension of preferences, anticipations, and concerns. This, in turn, empowers healthcare providers to offer more personalized assistance and guidance throughout the childbirth journey.

*"I think that it is the key to shared decision-making. It is a useful tool for meaningful conversations and to shine a light on both sides. And I think those who have the wish to, for example, not wanting something, I can tell them what I think from a medical point of view about it and from my own opinion, how I think about it as a healthcare provider which completes it. It is like, you see my side of the story and how can we both find a way to be satisfied? That is foremost important in shared decision-making because the view of the patient is most important, but that you are comfortable as the healthcare provider is just as important." (Participant 9-Healthcare provider)*

## The grounded theory: Birth plans as a tool for SDM

We created three key themes, namely "Creating a birth plan," "Getting all on board," and "Birth plans in daily decision-making practice," and offered valuable insights into the potential contribution of birth plans to SDM. Our findings suggest that the process of creating a birth plan serves as a stimulus, motivating women and their partners to engage in conversations to deliberate their preferences and gain a comprehensive understanding of various childbirth scenarios. This increased knowledge and preparedness of the women and partners, and enabled more effective and informed dialogues between women and partners, and between them and healthcare providers, regarding a broad spectrum of childbirth choices. Women, therefore, feel more in control and involved in their birth process and this fosters the ability to arrive at well-informed decisions. Furthermore, when healthcare providers review the jointly developed birth plan of the couple, it provides them with valuable insights into the preferences and needs of the individuals, thus facilitating their active participation in the SDM process and personalized care.

## Discussion

This study highlights the effectiveness of birth plans in enhancing SDM between pregnant women, their partners, and healthcare providers, emphasizing their contribution to both internal and external empowerment. Our grounded theory suggests that creating a birth plan encourages expectant women and partners to discuss preferences and gain knowledge about various childbirth scenarios, but it also enhances their ability to engage in effective conversations with each other and with their healthcare provider. Ultimately, this is perceived to lead to well-informed decisions through SDM [8,14,21–29].

Healthcare providers, women, and their partners all value SDM, acknowledging the birth plan as essential for guiding discussions on childbirth preferences. They concur on the necessity of realistic expectations due to childbirth's unpredictability. However, while women and their partners view the birth plan as a tool for empowerment and autonomy, healthcare

providers see it primarily as a means for communication and expectation management. Professionals aim to balance patient desires with medical advisability, ensuring safety and feasible outcomes. In contrast, expectant mothers and partners emphasize personal choices and control, highlighting the complex interplay between individual preferences, medical advice, and the uncertainties of childbirth.

Internal empowerment, as characterized by increased knowledge, confidence, and self-advocacy in women engaged in birth planning, is supported by multiple studies [10,29,30]. This empowerment reflects the psychological and emotional strength women gain from actively participating in their birth planning.

External empowerment is facilitated by the role of healthcare providers. The importance of healthcare providers' supportive and validating approach in the successful implementation of birth plans has been emphasized in the work of several researchers, highlighting how such support integrates women's preferences into clinical practice [31–33]. This necessitates clear communication and a deep respect for women's autonomous choices, integrating them seamlessly with the realities of medical practice.

The dialogue between healthcare providers and women, focusing on realistic expectations and the unpredictability of childbirth, is a critical component of external empowerment. Multiple studies have also noted the importance of these discussions in aligning expectations with feasible medical practices [22,27,32].

Further, the study underscores the necessity of ongoing education and communication about birth plans. Multiple studies advocate for antenatal education addressing care provision differences in various settings, assisting women in making informed choices regarding their birthing environment [8,31,34]. This aspect of education is crucial in ensuring that birth plans are both realistic and achievable.

Additionally, the recognition and respect for cultural and individual differences in birth plan creation and execution are paramount. This approach, as supported by research from authors such as Welsh et al., ensures that plans are tailored to individual needs, enhancing their effectiveness and contributing to a comprehensive approach to empowerment in SDM [31].

In conclusion, birth plans are instrumental in promoting empowerment and successful SDM. Their effectiveness depends on a balanced approach that respects medical guidelines, individual preferences, and cultural nuances, ultimately honoring the autonomy and preferences of the pregnant woman and her partner.

## Strengths & limitations

A strong point of this study is that it combined the perspectives of pregnant women, partners, and healthcare providers on the use of birth plans in facilitating SDM. It is important to study the perspectives of pregnant women, partners, and healthcare providers to understand the dynamics and potential barriers in the decision-making process.

Additionally, we ensured rigor of the findings by using purposive sampling, ensuring that the interviewers had no personal or work relationships with any of the participants, and a member check after the interviews.

There are some limitations to this study. One limitation is that it was conducted solely at Radboudumc. This is an academic teaching hospital providing secondary care for the local community and tertiary care for higher-risk pregnancies for the larger region. This hospital is committed to person-oriented care and the first in the Netherlands to endorse the Salzburg Declaration for shared decision-making [19]. This dedicated environment, already focused on collaborative care with patients, may have influenced the study's outcomes. However, this

focus also acts as a strength, providing a unique perspective on the efficacy of SDM within a pioneering healthcare setting dedicated to person-oriented and collaborative care. Secondly, the potential for recall bias should be considered, as the women, partners, and healthcare providers were interviewed one to six months postpartum. This time gap might have influenced their recollections, leading them to either ascribe more thought and meaning to their experiences or, conversely, to forget or downplay certain aspects of their experiences and decisions. This dual nature of recall bias could have impacted the results in varying ways. Additionally, healthcare providers are asked about their perspectives and opinions, not specific events or experiences. This may lead to only remembering thoughts and events that impacted them. This could lead to forgetting or not mentioning other events relevant to this study.

## Conclusion

This interview study explored how birth plans contribute to SDM in maternity care in the Netherlands, according to women, partners, and healthcare providers. The 30 interviews showed that most participants' experiences were positive. Women, partners and healthcare providers all value SDM, acknowledging the birth plan as essential for guiding discussions on childbirth preferences. Women and partners felt more prepared for birth by using the birth plan. The use of a birth plan seems to promote women's, partners', and healthcare providers' involvement in the birth process, and birth plans seem suitable to facilitate SDM as presented in our grounded theory. Multiple barriers to using birth plans for SDM have been discussed in this study, such as the attitude of healthcare providers toward a birth plan, or the expectations women and their partners have of the role of a birth plan. Therefore, more research is needed to investigate how to alleviate or circumvent these barriers.

## Supporting information

**S1 Table. Topic list interviews with women.**
(DOCX)

**S2 Table. Topic list interview with partners.**
(DOCX)

**S3 Table. Topic list interview healthcare providers.**
(DOCX)

**S4 Table. Characteristics of women participating in the interview study.**
(DOCX)

**S5 Table. Characteristics of women whose partners were interviewed in the study.**
(DOCX)

**S6 Table. Characteristics of healthcare providers participating in the interview study.**
(DOCX)

## Acknowledgments

We would like to express our gratitude to the women, partners, and healthcare providers who contributed to this research through their valuable insights shared during interviews

We acknowledge and thank Naomi Scholten for conducting the interviews with the women included in this study, which greatly contributed to the quality of the data. We would also like to express our gratitude to dr. Martine Hollander, a gynecologist, for her assistance and support during the interviews.

## Author Contributions

**Conceptualization:** Naaz Shareef.

**Data curation:** Naaz Shareef.

**Formal analysis:** Naaz Shareef, Silke Lamers.

**Investigation:** Naaz Shareef, Poshya Said, Silke Lamers.

**Methodology:** Naaz Shareef, Poshya Said, Silke Lamers.

**Project administration:** Marianne Nieuwenhuijze, Marieke de Vries, Jeroen van Dillen.

**Resources:** Naaz Shareef, Poshya Said.

**Supervision:** Marianne Nieuwenhuijze, Marieke de Vries, Jeroen van Dillen.

**Visualization:** Naaz Shareef.

**Writing – original draft:** Naaz Shareef, Poshya Said, Silke Lamers.

**Writing – review & editing:** Naaz Shareef, Marianne Nieuwenhuijze, Marieke de Vries, Jeroen van Dillen.

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
