## [Decision Letter · Decision Letter 0]

2 Jan 2024

PONE-D-23-38329The contribution of birth plans to shared decision-making from the perspectives of women, their partners and their healthcare providersPLOS ONE

Dear Dr. Shareef,

Thank you for submitting your manuscript to PLOS ONE. After careful consideration, we feel that it has merit but does not fully meet PLOS ONE’s publication criteria as it currently stands. Therefore, we invite you to submit a revised version of the manuscript that addresses the points raised during the review process.

You will find the comments from the reviewers below. 

We look forward to receiving your revised manuscript.

Kind regards,

Ivan Sarmiento

Academic Editor

PLOS ONE

3. PLOS requires an ORCID iD for the corresponding author in Editorial Manager on papers submitted after December 6th, 2016. Please ensure that you have an ORCID iD and that it is validated in Editorial Manager. To do this, go to ‘Update my Information’ (in the upper left-hand corner of the main menu), and click on the Fetch/Validate link next to the ORCID field. This will take you to the ORCID site and allow you to create a new iD or authenticate a pre-existing iD in Editorial Manager. Please see the following video for instructions on linking an ORCID iD to your Editorial Manager account: " ext-link-type="uri" xlink:type="simple">https://www.youtube.com/watch?v=_xcclfuvtxQ".

Reviewers' comments:

Reviewer's Responses to Questions

**Comments to the Author**

1. Is the manuscript technically sound, and do the data support the conclusions?

Reviewer #1: Yes

Reviewer #2: Partly

2. Has the statistical analysis been performed appropriately and rigorously? 

Reviewer #1: N/A

Reviewer #2: N/A

3. Have the authors made all data underlying the findings in their manuscript fully available?

Reviewer #1: Yes

Reviewer #2: Yes

4. Is the manuscript presented in an intelligible fashion and written in standard English?

Reviewer #1: Yes

Reviewer #2: Yes

5. Review Comments to the Author

Reviewer #1: Overall Impression

This manuscript describes how birth plans can be used by women, their partners and healthcare providers to facilitate shared decision-making and pertinent communication during pregnancy and birth. Having the perspectives of the pregnant women, of partners and healthcare professionals adds to the understanding of the issue. It also emphasizes important barriers on the use of birth plans from various angles.

The design of the study, sampling strategy and data analysis are appropriate and described in detail. The Themes and Sub-Themes are easy to follow and the quotes are pertinent. The Results contribute to obstetrical knowledge and are transferable to other practice settings.

A few issues could improve the overall understanding of the topic, as well as the readability of the manuscript.

Areas for improvement

1. It would be useful to have one sentence about the Netherlands’ context and how pregnant women are usually followed (primary/secondary follow-up/referrals during pregnancy or birth) as systems vary from one country to another.

2. The paragraph in the literature review referring to the aims of a birth plan lacks references.

- A 2022 systematic, integrative review on Birth plans by Bell et al, offers a variety of references that could be used.

3. In the Methods paragraph, the following sentence should be clarified to avoid confusion: “We combined data from three interview studies concerning birth plans in relation to SDM from women’s, partners’, and healthcare providers’ points of view.” It could be interpreted as data from 3 different studies were merged, whereas 3 groups of people were interviewed within one study.

4. Some paragraphs could gain readability by making them more concise and ensuring there are no redundant sentences. For example:

a. The paragraph on the Research Team and Reflexivity. Some sentences could be removed.

b. The grounded theory: birth plans as a tool for SDM. Many points from the Discussion are repeated.

c. The discussion itself could be more concise.

d. In the paragraph on Strengths and Limitations, the purpose of the lengthy description of the hospital is unclear.

5. Some sentences could be reworded for the article to gain readability. Some examples:

a. “Given the importance of SDM and the possible impact of birth plans, research is needed to gain more insight into how this can work.”

b. “The interviews brought to the attention that all stakeholders found it important that they talked to each other about their preferences for birth, while at the same time remaining realistic in making decisions about their birth and taking in mind that not all options may always be possible or desirable, because they might consider some risks to be unacceptable for the woman or her baby.”

c. “Women expressed the anticipation of having choices”

6. Some sentences are incomplete. For example:

a. “The knowledge that certain aspects (such as fulfilling all the preferences outlined in the birth plan, which may be impossible) adds more pressure to their role.”

Reviewer #2: I consider the document presents an interesting work that particularly contributes to the research field showing how women, their partners and health providers perceive the contribution of birth plans to shared decision-making (SDM). Methodologically, the document describes in a sufficient and adequate form the process of data collection and analysis. The research results are presented in an orderly manner and are consistent with the methodology used.

According to what the authors and other research have also found, the use of birth plans seems to contribute to SDM. Nevertheless, some difficulties have also been identified. Among the findings of this research, that describes what three groups of stakeholders say about the topic, some characteristics that can contribute to solving the problems have been described.

The document also has some aspects that I consider the authors must improve to clarify, organize, and consolidate the work. By section these are my recommendations:

Introduction

The authors present the Shared Decision-Making (SDM) part in a clear and precise way. The information related to de aims of a birth plan however is not clearly supported. The reference 8 is a study made in women with high-risk pregnancies, and the reference 11 is a link to a Dutch web page where it is not possible to verify the cited information in English.

This section lacks in my opinion information about the context where the research was done. For example, the childbirth care in the Netherlands has some particularities that would be important to mention.

Methods

This section is one of the strongest parts of the paper.

In this section the authors included data that I consider must be presented in results. For example, in the subtitles Women and partners and Healthcare providers, they present numbers of contacted persons. They also cite tables (S1, S2 and S3) that I supposed will present data of the contacted persons, but instead they present information of the interviewed. Besides, the title of Tables S2 is not precise; most of the information included in this table it´s not related to partners.

The authors also insert the Figure 1 in the methods section, but the categories it presents are a result of their analysis.

In the first part of this section the information about the institution where the research was done is missing; some information is presented in the Strengths limitations section. Besides it would give the readers a better perspective, it would contribute to understand the section 3.1 of Results, which is directly related to some of the characteristics of the institution.

I also recommend the authors to explain with more detail the process and the criteria based on which the participants have been chosen. For example, it´s not clearly described how they select the interviewed among the approached; neither which is the relationship between each women and the partners and health personal interviewed.

Results

I consider this section the best part of the paper.

In the first part of it, I recommend the authors to present a more detailed view of the general characteristics of the participants.

There is an error in the number of the tables cited at the end of the first paragraph (the correct ones are S1, S2 and S3).

Discussion

As the authors highlight, I also consider the principal contribution of their work is the triple perspective (women, their partners and, health providers) of the research topic, but the discussion is focused on the findings of each group in a separate form. I consider the authors should highlight the similarities and differences in the way each group perceives the problem and a how they can complement or contradict each other. This analysis can contribute to identify and possibly help, to resolve the problems described in the use of birth plans in SDM.

References

I found many errors and inconsistencies in this section.

On page 31 a paper from Fernandez-Arroyo is cited, but it is not included in the references. The author´s last name cited in page 33 (Farahat) is incorrect. The reference related to COREQ (25), it is not included in the final references.

In this references the central topic discussed in the cited paper is not clearly related to the topic discussed by the authors: 15 do not assess birth plans; 19 is about women’s motivations for choosing a high-risk birth setting; 20 is about traumatic childbirth and birth plans were no assessed; 24 is a research on the use of birth plans in Life-Limiting Fetal Diagnoses; 27 is a study to validate the reliability of a Farsi version of a scale use to measure different aspects of childbirth fear; 29 is a paper about the concept of a woman's empowerment in pregnancy, childbirth and the postnatal period; 30 is a research of midwives perceptions of their role within the context of maternity service reform.

Reference 26 is not traceable.

References 36 and 37 are web pages in Dutch. It is not possible to verify the cited information in English.

Strengths limitations

The authors should explain better the reasons why the Radboudumc care characteristics are considered a limitation for the study. Also, I consider the women recall bias can influence the results not only in the “more thought and meaning” way, but also in the opposite way.

The form that the authors use to describe how the findings of other studies are related to their own findings frequently is not clear. For example, they present their findings and then insert some citation numbers, without any comments.

6. PLOS authors have the option to publish the peer review history of their article (what does this mean?). If published, this will include your full peer review and any attached files.

Reviewer #1: No

Reviewer #2: **Yes: **Andrés Cañón

---

## [Author Response · Author response to Decision Letter 0]

7 Mar 2024

Dear Reviewers,

I am writing in response to the comments received for our manuscript submission, with the submission number PONE-D-23-38329. First and foremost, I would like to express my sincere gratitude to the reviewers for their insightful feedback and constructive suggestions. Their expertise and thoughtful critique have been invaluable in enhancing the quality and clarity of our work.

In response to the specific comments and suggestions provided, we have made the following revisions and improvements to our manuscript, as detailed in our document titled "Response to Reviewers":

We believe that these revisions have significantly improved our manuscript, making it a stronger and more valuable contribution to the field. We appreciate the opportunity to revise our work based on the reviewers' insightful feedback and hope that the changes meet their expectations and the high standards of the journal.

Thank you once again for the opportunity to refine our manuscript. We look forward to any further comments or suggestions you may have and are eager to see our work contribute to the broader academic and scientific community.

Sincerely,

Naaz Shareef

---

## [Decision Letter · Decision Letter 1]

16 Apr 2024

PONE-D-23-38329R1The contribution of birth plans to shared decision-making from the perspectives of women, their partners and their healthcare providersPLOS ONE

Dear Dr. Shareef,

Thank you for submitting your manuscript to PLOS ONE. After careful consideration, we feel that it has merit but does not fully meet PLOS ONE’s publication criteria as it currently stands. Therefore, we invite you to submit a revised version of the manuscript that addresses the points raised during the review process.

**Kindly take into consideration the recommendations of Reviewer #1.**

We look forward to receiving your revised manuscript.

Kind regards,

Ivan Sarmiento

Academic Editor

PLOS ONE

Journal Requirements:

Reviewers' comments:

Reviewer's Responses to Questions

**Comments to the Author**

1. If the authors have adequately addressed your comments raised in a previous round of review and you feel that this manuscript is now acceptable for publication, you may indicate that here to bypass the “Comments to the Author” section, enter your conflict of interest statement in the “Confidential to Editor” section, and submit your "Accept" recommendation.

Reviewer #1: (No Response)

Reviewer #2: (No Response)

2. Is the manuscript technically sound, and do the data support the conclusions?

Reviewer #1: Yes

Reviewer #2: (No Response)

3. Has the statistical analysis been performed appropriately and rigorously? 

Reviewer #1: N/A

Reviewer #2: (No Response)

4. Have the authors made all data underlying the findings in their manuscript fully available?

Reviewer #1: Yes

Reviewer #2: (No Response)

5. Is the manuscript presented in an intelligible fashion and written in standard English?

Reviewer #1: Yes

Reviewer #2: (No Response)

6. Review Comments to the Author

**Reviewer #1: **Thank you for kindly taking my comments into consideration.

The description of the Netherlands context was very clear and the readability of the sections mentioned greatly improved.

I would like to suggest some minor recommendations, mostly grammatical, before publication:

Abstract :

- Barriers are the attitude of healthcare providers toward birth plans, such as their openness resistance (page 1 and line 44)

Consider “evident resistance”? as openness resistance does not make sense.

Methods

- Line 112: “These studies”. Do you mean “These interviews”?

- Lines 114 and 597: This is «an» academic

- Lines 117 and 600: Please reference the Salzburg Declaration for shared decision-making

- Lines 125 and 141: remove “-“ after information

- Line 140: add space before “All partners”

- Lines 141-2: “Only after agreeing with the consent form were they included in the study”

- Lines 159: “with the women or partners” (since it was not the women’s partners

- Line 169-170: to “ensure the information in the interviews was correct”

- Line 191: can be found in Fig 3.

Results

- Line 145: remove the first “actively”

- Lines 258-59: “expectations, which will be further elaborated in the…” – otherwise the second sentence is incomplete.

- Lines 334: remove “either”

- Line 367: the information goes “in one ear and out the other”

- Lines 446-450 are saying the exact same thing as Lines 451-456. Please choose.

- Line 461: “I know this”

- Line 527: the sentence sounds incomplete, perhaps add «and» before «offered»?

Discussion

- Line 602: the efficacy «of» SDM

Thank you.

**Reviewer #2:** (No Response)

7. PLOS authors have the option to publish the peer review history of their article (what does this mean?). If published, this will include your full peer review and any attached files.

Reviewer #1: No

Reviewer #2: No

---

## [Author Response · Author response to Decision Letter 1]

20 May 2024

Dear Dr. Ivan Sarmiento,

We sincerely thank you and the reviewer for carefully reading our manuscript and for the constructive feedback provided. We appreciate the time and effort you took to review our work.

We have made all the suggested minor revisions, primarily grammatical, in accordance with your recommendations. We have clearly indicated each correction and modification in the attached document using track changes to facilitate your review.

We believe that these changes have improved the clarity and readability of our manuscript and hope that it now meets the standards for publication in PLOS ONE. We look forward to your final decision.

Thank you for your valuable input and consideration.

Sincerely,

Naaz Shareef

On behalf of my research team

---

## [Editor Report · Decision Letter 2]

28 May 2024

The contribution of birth plans to shared decision-making from the perspectives of women, their partners and their healthcare providers

PONE-D-23-38329R2

Dear Dr. Shareef,

We’re pleased to inform you that your manuscript has been judged scientifically suitable for publication and will be formally accepted for publication once it meets all outstanding technical requirements.

Kind regards,

Ivan Sarmiento

Academic Editor

PLOS ONE

Additional Editor Comments (optional):

**Please consider acknowledging as a limitation that the study only included women with healthy pregnancies and childbirths, so the experiences of women with maternal health issues are unknown.**
---

## [Editor Report · Acceptance letter]

3 Jun 2024

PONE-D-23-38329R2 

PLOS ONE

Dear Dr. Shareef, 

I'm pleased to inform you that your manuscript has been deemed suitable for publication in PLOS ONE. Congratulations! Your manuscript is now being handed over to our production team.

Kind regards, 

on behalf of

Dr. Ivan Sarmiento 

Academic Editor

PLOS ONE